# Mechanisms and Health Aspects of Food Adulteration: A Comprehensive Review

**DOI:** 10.3390/foods12010199

**Published:** 2023-01-02

**Authors:** Mysha Momtaz, Saniya Yesmin Bubli, Mohidus Samad Khan

**Affiliations:** Department of Chemical Engineering, Bangladesh University of Engineering & Technology (BUET), Dhaka 1000, Bangladesh

**Keywords:** food adulteration, physical properties, food substitution, synthetic adulterants, health hazard

## Abstract

Food adulteration refers to the alteration of food quality that takes place deliberately. It includes the addition of ingredients to modify different properties of food products for economic advantage. Color, appearance, taste, weight, volume, and shelf life are such food properties. Substitution of food or its nutritional content is also accomplished to spark the apparent quality. Substitution with species, protein content, fat content, or plant ingredients are major forms of food substitution. Origin misrepresentation of food is often practiced to increase the market demand of food. Organic and synthetic compounds are added to ensure a rapid effect on the human body. Adulterated food products are responsible for mild to severe health impacts as well as financial damage. Diarrhea, nausea, allergic reaction, diabetes, cardiovascular disease, etc., are frequently observed illnesses upon consumption of adulterated food. Some adulterants have shown carcinogenic, clastogenic, and genotoxic properties. This review article discusses different forms of food adulteration. The health impacts also have been documented in brief.

## 1. Introduction

Food are organic substances consumed for energy, growth, and nutritional purpose. Food adulteration refers to the process through which the quality of food is lowered [1,2]. Broadly, food adulteration is a category of food fraud which is accomplished deliberately by human beings for financial gain [1,2,3,4,5]. It is also termed as economically motivated adulteration (EMA) that sometimes gives rise to authenticity issues: brand, origin, manufacturing ingredients, and their composition are often misrepresented [6]. Protected Designation of Origin (PDO), Protected Geographical Indication (PGI), Certificate of Specific Character (CSC), and Traditional Specialties Guaranteed (TSG) are some of the familiar terms [7]. Adulterated food products pose several health hazards, including health diseases, and they weaken the immune system. 

Adulteration of foodstuffs has been frequently observed for centuries due to the contribution of several common reasons. The perishable nature, heterogeneity, and huge production of certain food items have always been tempting for dishonest traders; the similarity and diversity of animal species, stock limitation, and market price pressure also encourage them to perform intentional adulteration [8]. Some food items have been severely prone to adulteration due to possessing high dietary value and vast popularity. Food with a narrow profit margin also have frequently appeared in fraud lists [9,10]. Beverages and other liquid foods have drawn special attraction, with a wide variation in chemical composition, high quality, long aging time, and high production cost [11,12]. The competitive nature of the food industry due to consumer’s extensive demand for variety and low-cost food products has stimulated this issue further. In addition, the limitation of raw materials, demand–supply gap, and the ever-present tendency to reduce cost and maximize profits have created more opportunities and interest for invidious traders [13,14]. Other reasons include degraded moral society, spoiled socio-economic structure, and low legal standards, and their improper enforcement may play a significant role [15,16]. 

Many food adulteration incidents were encountered in the past. The China gutter oil scandal is such an example that used illicit cooking oil from restaurant fryers, grease traps, and slaughterhouse waste, or extracted from discarded animal parts. Later, such contaminated food products were also found in Taiwan, Hong Kong, and Singapore. Addition of Sudan dye was reported in fats and oils, herbs and spices, food additives, and flavorings several times. In 2007, two cases of tetrodotoxin poisoning were found, which were caused by substitution of monkfish with pufferfish. In 2008, the contamination of powdered infant milk in China caused illness in 294,000 children, with 50,000 hospitalizations and 6 deaths. In 2013, a methanol poisoning affected 694 patients, with 8 deaths in Iran. In 2018, methanol toxication caused extensive hemorrhagic cerebral infarction or multiple organ failure that affected 90 individuals, with 64 fatalities in Malaysia [17]. Dioxins in pork in 2008, milk with detergent, fat, and urea in 2012, and processed beef products with horsemeat in 2013 are also some renown incidences [18]. It is reported that the global food industry faces an expense of approximately USD 10-15 billion per year due to health and financial damage associated to intentionally contaminated food products [19]. The demand for food increased with the emergence of COVID-19 and Brexit, which resulted in reduced industry inspections, weakened governance, audits, and ever-increasing pressure on the food industry [20].

Several incidents on food adulteration were reported in recent times. The Trello and European Commission food fraud databases reveal many such incidents. As part of Operation OPSON XI, coordinated by Europol for EU-wide action and which took place between December 2021 and May 2022, the authorities of 26 countries seized almost 27,000 tons of fake food and 15 million liters of alcoholic beverages. In October 2022, Food Safety and Halal Food Authority of Pakistan seized 4000 L of fake honey adulterated with sugars, chemicals, and wax. In December 2022, more than 12 people died and more than 15 people lost their eyesight in Bihar, India, after consuming adulterated alcohol. In April 2022, the Spanish Agency for Food Safety and Nutrition launched an alert regarding the production of fake olive oil and extra-virgin olive oil adulterated with other vegetable oils. In Turkey, it was reported in August 2022 that ground sumac was found to be adulterated with unauthorized color. In August 2022, curry powder from Cameroon was found adulterated with Orange II by authorities in Belgium. In August 2022, authorities in Pakistan seized tea made with textile dyes, sawdust, and other fillers. Cumin seeds mixed with 30 tons of adulterants were seized by authorities in India in July 2022. Between January 2022 and March 2022, the FDA collected and tested 144 samples of imported honey and found ten percent of the samples to be adulterated with undeclared added sweeteners. In Germany, sausage and poultry meat were alleged to contain undeclared “mechanically separated meat” ingredients (March 2022). 

The technical progress for food adulteration research is quite prominent in modern times. Earlier, adulterated food products were identified on the basis of a few physical parameters, such as refractive index, viscosity, melting point, saponification, iodine value, etc. With the expansion of global markets and business competition, the frequency of food adulteration has increased exponentially, which gave rise to the necessity of highly efficient techniques. The food governing authorities around the world have also established the official methods for detection of food adulteration. Currently, chromatography and spectrometry are widely used analytical techniques. Protein and DNA-based techniques are also in practice. In addition, metabolomics, hyperspectral imaging, and chemometrics are some other techniques. The state-of-the-art techniques are highly contributing to combat food adulteration. Still, there remain some drawbacks, such as complexity, excessive use of toxic compounds, laborious sample preparation, etc. [21,22]. Lack of collaboration between scientists and food policy-makers might be an obstacle in identifying the need for proper research and development. In addition, there lies a gap between laboratory research and practical aspects [23]. Food adulteration is a complicated phenomenon and involves the availability of numerous fraudulent options discovered by dishonest people. It is important to understand different forms and the dynamic nature of food adulteration and their impacts on human health. Regular market studies and the development of laboratory techniques for the qualitative and quantitative assessment of new forms of food adulterants would be critical. Gathering and analyzing the possible forms of adulteration in a single frame would be highly useful to bridge the gap for research. 

This article presents a cohesive review of the common forms of food adulteration and origin fraud of food. In this article, the health impacts associated with adulteration have also been addressed in brief. The scientific literature for the period of 1995 to 2022 was searched using the keywords “food adulteration”, “economically motivated adulteration”, “origin fraud”,” mislabeling”, and “health impact of food adulteration”. Approximately 400 scientific articles and reports were screened and reviewed. Of those articles, 174 were found more relevant and reviewed extensively. The Trello and European Commission food fraud databases along with relevant newspapers and magazine articles were reviewed for recent food adulteration incidents. The information covered in this article will help researchers, food engineers, scientists, and policy-makers to combat food adulteration and health hazard.

## 2. Overview of Food Adulteration

The main purpose of food adulteration is to alter the quality of food products for economic advantage. Such actions usually take place by substitution with inferior quality or less valued food and increasing the weight or volume by admixture of undeclared ingredients. Providing a more attractive appearance by injection of artificial chemicals and colorants is also included here [8,24].

The principal aim behind food substitution is to reduce raw material and manufacturing cost by incorporation of inferior compounds. The physical properties and taste of food items are modified in many ways. Artificial ripening of fruits is a common means of food adulteration. Adulteration with preservatives, colorants, and artificial sweeteners are also common food adulteration techniques. Among other adulterations, falsification of origin is also labeled as adulteration, as it includes false claim for superior origin. 

Figure 1 presents the key forms of intentional food adulteration, which are briefly discussed in the following sections [3,4,5].

## 3. Adulteration to Improve Physical and Sensory Properties

Taste and appearance have a high impact on the commercial value of food products. Increasing the shelf life of food items gives financial benefits. Artificial ripening and sweetening are used to increase food palatability. Similarly, artificial colorants are applied to improve the appearance of food. Preservatives are added to store food in a fresh condition for a long time. The following sections briefly describe different means of adulteration to improve physical and sensory properties.

### 3.1. Artificial Ripening Agents

To avoid the economic loss incurred by the spoilage of climacteric fruits during the harvest, processing, and transportation process, the fruit sellers pluck the fruits much before they attain proper maturity and use chemicals to ripen them artificially just before retailing. The necessity for artificial ripening also arises if the fruit sellers wish to sell fruits before their due season to make additional profit. Ethylene, ethanol, methanol, propylene, methyl jasmonate, ethylene glycol, ethephon, and calcium carbide are used to ripen fruits and vegetables artificially [25,26,27,28,29,30,31]. The ripening agents and their key features are documented in Table 1.

### 3.2. Growth Hormones

Gibberellic acid, alpha naphthyl acetic acid, and oxytocin are growth hormones used on fruits and vegetables by farmers to trigger growth. Oxytocin, being a mammalian hormone and a veterinary drug, is not suitable for vegetable crops. However, it is widely used in bottle gourds, bitter gourds, pumpkins, and cucumbers to enhance size and color [29].

### 3.3. Artificial Sweetening Agents

Sweetness of food is an important criterion in terms of demand and marketability. So, there is always a tendency by the traders to increase the sweetness of selected food items by artificial means. Mainly, fruits and vegetables, beverages, sweeteners, and confectionary products fall into sweet food category.

Artificial sweeteners are injected by injector pumps on one side of fruit to alter the natural sweetness. To ensure uniform distribution, sellers inject sweeteners from several points in the fruit. Saccharine mixture was found to be injected into melons and watermelons to enhance sweetness artificially [29]. The addition of external sugar or sugar solution is a common form of adulteration of fruit juice. In the case of export purpose, the concentrated fruit juice is shipped; later, external water and sugar are added to the concentrated fruit juice to give natural properties similar to that of natural juice [11]. In addition, high-fructose corn syrup, partially hydrolyzed cane syrup, and beet medium invert sugar are also added to increase the Brix value and to improve composition quite similar to authentic juice [36]. The direct addition of sucrose, glucose, fructose, maltose, beet sugar, corn sugar, and cane sugar in honey is widely practiced [37]. Adulteration of honey and confectionary products with fructose or glucose changes the fructose-to-glucose ratio [38]. Feeding honeybees with syrups and industrial sugars after the broods have been naturally available is known as an indirect form of adulteration which is very difficult to detect. Feeding low quality honey to honeybees is also reported [38,39,40,41]. Different confectionary and bakery products are sweetened with Acesulfame-K and Aspartame [42,43]. Sugar, coloring agents, synthetic red dyes, aromatizing agents, and sweeter foreign wines are often added into wine for quality enhancement purpose [44,45]. Glycerol reduces the acidity and bitter taste, increases the sweetness, and stops fermentation, while diethylene glycol imparts relish to wines. The addition of root or cane sugar to tequila and the addition of cane or beet ethanol to whiskey have also been noted [46,47].

### 3.4. Artificial Coloring Agents

Color, texture, and appearance of food products are important criteria in the selection of desired food items by consumers. Foods with attractive color increase marketability and profit. Thus, following this trend, various natural as well as synthetic dyes are applied to different food items. Most of the colorful food items are at risk of such malpractice. Among those, fruits and vegetables, egg and egg-derived food, spices, sweeteners, and confectionary products are prime choices. Coloring agents are also added in processed meat and fish [48]. Paprika oleoresin is commonly used as a natural coloring agent in meat-based food products [49].

Natural colors, such as chlorophyll, annatto, and caramel, have been reported several times in fruits and vegetables. Synthetic dyes are more popular among the sellers, as those exceed natural ones in many aspects. These are chemically synthesized, more stable, shiny, and highly efficient. Their cheapness and easy availability are also considerable factors. Rhodamine B, auramine, Metanil yellow, Congo red, Orange II, malachite green, and other permitted and non-permitted colors are used in cut fruits and vegetables. Red dye is injected into watermelons to enhance the acceptability of the consumers. Malachite green is widely used to make green vegetables, such as green chili, green peas, bitter gourds, lady finger, and pointed gourdd, look greener, fresh, bright, and glowing. Another reagent frequently reported for bitter gourd and lady finger to be dipped into is copper sulfate solution, which is bright blue in hydrous form and pale green in anhydrous form. Mobile oil for coloring and carbofuran for a fresh purple appearance are injected into brinjals, tomatoes, cauliflowers, and cabbages. Phosphomidone, methyl parathion, monocrotophos, and formaldehyde are also injected for a fresh white appearance [29]. 

Eggs are dyed artificially by azo dyes. The addition of illegal synthetic dyes is inspiring among the traders, as eggs’ nutritional value and freshness are predicted by judging the egg yolk color. Sudan dyes are a type of synthetic azo dye that is used in industry or printing. Another azo dye, Para red, is chemically similar to Sudan dye I and is used for printing purposes. The yellow-orange-hue-colored eggs are preferable by consumers. So, the sellers often feed hens food mixed with dyes to enhance the color of egg yolk [50].

Sudan I, Sudan Ⅳ, Metanil Yellow, Sudan III, Oil Orange SS, Rhodamine B, Auramine O, coal tar red, Para Red, etc., are applied in red pepper chili powder. Sudan I, Sudan Ⅳ, Acid Black I, Annatto, etc., are mixed with paprika powder. Sudan I, Metanil Yellow, Lead Chromate, etc., are added with turmeric powder. Amaranth Red and Basic Red 46 are added into sumac. Auramine O and Chrysoidin are mixed with curry powder. Acid Orange II, Metanil Yellow, Ponceau 4R, Gardenia Yellow, dye extracted from the flowers of Buddleja officinalis Maxim, etc., are added with saffron flower. Crystal Violet is mixed into cayenne pepper. In addition to synthetic dyes, colored paper and wood are also used [9].

Dark honey or “forest honey” is richer in minerals and nutritional value than light honey and possesses a higher commercial value. That is why light honey is often tainted with sulfite ammonia caramel and presented as dark honey [40,41].

### 3.5. Adulteration to Improve Food Consistency

Detergents, along with oil and fat, have been used to improve consistency of dairy products. In addition, one of the reasons for the accidental presence of detergent in milk products is lack of hygiene, sanitation, and improper cleaning in the dairy industries [51,52,53]. Along with detergent, other compounds, such as salt, glucose, starch, neutralizers, pulverized soap, surfactant, and coloring agent, are also mixed to adjust the whiteness, viscosity, thickness, and solid-not-fat content. Addition of gelatin, stabilizers, enzymes, and external reagent to improve food consistency are also reported [15,51]. 

### 3.6. Preservatives

Preservatives are usually added to increase the shelf life of food items. Addition of preservatives is one of the most practiced forms of adulteration at present. Fruits and vegetables, fish and seafood, meat and processed meat, milk and dairy products, and beverages are the most tempting targets.

Formaldehyde is the most reported preservative used in fruits and vegetables. Formalin inhibits the growth of microbes by interacting with the amino groups of adenine, cytosine, and guanine and denaturing them. It also penetrates the interiors of bacterial spores, which makes it capable of preventing microbial contamination and prolonging the shelf life of food. Although there is no set standard for the daily intake of formaldehyde from food, the World Health Organization (WHO) estimated it to be in the range of 1.5–14 mg/d (mean 7.75 mg/d) for an average adult. According to the European Food Safety Authority (EFSA), the daily oral exposure to formaldehyde should not exceed 100 mg formaldehyde per day [54]. In a recent study, the formaldehyde level in fish and seafood were found to be higher than the recognized safety level of 5 mg/kg [55].

In addition to formalin, wax components, and different forms of esters, are also used to reduce the water loss and surface abrasion. The other purpose is to control internal gas composition and provide the shiny appearance of those fruits and vegetables that lost their natural wax during primary processing. Compared with other lipid and non-lipid coatings, wax coatings provide better resistance to moisture loss. Due to petroleum-based wax containing harmful wood rosins and solvent residues, beeswax, carnauba wax, and shellac are preferable. Fruits and vegetables are dried well before waxing and handling [29]. With the increasing demand and distant transportation of fish and seafood, the addition of low-cost preservatives has been a long-practiced issue. Fish, dry fish, and seafood are usually adulterated with preservatives, such as formalin, chlorofluorocarbon, and DDT powder, to tackle the spoilage and quality deterioration [56]. External parasites of fish egg are also treated with formalin [57]. Since a small quantity of formaldehyde is also naturally produced in fish as a byproduct of amine oxide degradation, artificially added formalin is quite difficult to detect [58]. Increment of shelf life by inhibiting microbial activities is a common form of milk and dairy product adulteration. It is accomplished by adding several preservative chemicals, such as formalin, urea, nitrate or pond water, borax acid, boric acid, cane sugar, sucrose, glucose, caustic soda, salicylic acid, hydrogen peroxide, benzoic acid, hypochlorite, and potassium dichromate [15,59,60]. Hydrogen peroxide acts as a preservative for pasteurized milk by activating the natural enzyme lactoperoxidase [60]. On the other hand, a significant reason for adding urea in dairy products is to elongate the shelf life. Preservatives, such as salicylic acid and benzoic acid, are added for preserving cheap wines prone to souring. Citric acid is added for pH adjustment. Beet sugar, cane sugar, concentrated rectified must, grape must, or grape wine are added to increase the natural content of ethanol and overall commercial value [51].

The key preservatives used in food items are listed in Table 2.

## 4. Food Substitution 

Substitution is the most diverse form of intentional food adulteration, which includes the direct alteration of a part or whole food item or external addition of other inferior food products or fake nutritional compounds. The common forms of food substitution are discussed in the following section.

### 4.1. Substitution of Species and Tissue

#### 4.1.1. Fishery Substitution:

Substitution of fish species is the most practiced form of fish fraud [63]. It refers to the replacement of a highly reputed fish species by a bad or inferior species [14]. A commercially valuable fish species is substituted with a low-value, non-declared, and non-specified species to make extra profit and to compensate for high tariffs paid for some species [14,19,64]. Crustaceans and high-quality shrimps are more prone to substitution due to their high market demand [65]. The traditional method of species identification is morphological analysis; however, in the case of seafood, it is quite inefficient as those are phenotypically similar and their external body parts are often removed during processing [63]. A risk of willful or unintentional substitution also prevails as visual specification becomes more difficult once fish has been processed into another form [66].

#### 4.1.2. Substitution in Meat Products

In the case of inter-species substitution, meats with similar color, such as beef and horse meat, beef and mutton, or poultry and pork are visually quite difficult to distinguish when frozen or processed into another form and shape. Sausage is one such processed meat product highly relished worldwide that is traditionally made from intestine or obtained synthetically. Though it can be made from beef, chicken, or pork, fraudulent substitution of species is also prevalent here [67]. Minced meat, one of the most versatile meat products, used in hamburgers, patties, meatballs, sausages, and salami, is prone to frequent adulteration by substitution with other meat species [68]. In this case, identification of meat species is very difficult as the morphological structure gets removed during mincing, and the adulterated minced meat appears very similar to the authentic product [69]. The undeclared addition of animal tissue, such as collagen and offal, are also prevalent, which is profitable for the traders [70,71]. Due to pork being cheaper and more readily available than other meat species, frequent substitution of other meat with porcine meat is reported [67,72,73]. 

#### 4.1.3. Substitution in Milk Products

The milk used to manufacture dairy products are usually derived from cow, sheep, goat, and buffalo. Cow milk is widely used both in developed and developing countries; however, there are incidents when cow milk is avoided due to allergenic reaction, religious restriction, ethical or cultural issues, personal preference, and impudence for certain food products [74,75]. Fraudulent substitution of other expensive milk with cow milk is very common. She-donkey milk, possessing high commercial value, is often substituted by cow or goat milk [60]. People with allergenic problems to cow milk prefer caprine milk due to its being easily digestible and containing low lactose content. Another example of malpractice is representing bovine milk as caprine milk [74,76]. Fraudulent replacement of sheep and goat milk with cheaper cow milk is also reported [77]. Replacement of fresh cow milk with reconstituted skim milk powder due to the popularity of cow milk is also reported [78].

Traditional cheese products, such as feta, manchego, and pecorino are manufactured from the mixture of ovine and caprine milk or solely ovine milk. However, the seasonal production and higher price of ovine and caprine milk make those cheese items prone to substitution by cheap bovine milk. Cheese products labelled as “pure buffalo mozzarella” are often found to contain cow milk. Cheese products made from one pure species and with protected designation of origin are rarely proved authentic and should be monitored from time to time [77]. 

#### 4.1.4. Coffee Substitution

Admixture of costly coffee beans with comparatively cheap beans is a very common practice. The two commercially important species, Arabica coffee (*Coffea arabica*) and Robusta coffee (*Coffea canephora*) are different in quality and botanical characteristics. Arabica coffee is costlier and has greater acceptability than robusta coffee due to its organoleptic properties. As a blend of these two varieties is also available in the market, the valid representation and labelling as pure Arabica is often susceptible. It is possible to distinguish these two types of beans through visual inspection; however, for coffee in ground and roasted form, it is not possible to distinguish in such a way [79,80,81]. Similarly, other commercially valued high quality and rare coffee species, such as Kona coffee grown in Kona Island, Blue Mountain, Tanzanian Peaberry, and Indonesian palm civet coffee (*Kopi luwak*), are often substituted by coffee beans with cheap species. Foreign fillers and coffee byproducts are also admixed with pure coffee products [79].

### 4.2. Substitution of Protein Content

#### 4.2.1. Protein Substitution in Meat

Animal protein, such as egg, gluten, and porcine gelatin are often added to meat products to increase the apparent protein content [67]. Soybean proteins and cereal flours are used in sausages to recover the desired all-meat properties, such as emulsifying capacity, emulsion stability, and water-binding capacity. Plasma protein is a complex mixture of serum albumin, globulins, and fibrinogen that possesses the ability to produce and stabilize foams and emulsions and to form heat-induced gel. It is utilized by the food industry to control the texture and desired structure of processed meat products [82,83]. Enzymes such as fibrinogen and thrombin are used as blood-based binding agents both in chilled and raw states to give a desired mass and shape to processed meat products [71,82]. Thrombin is used in conjunction with blood plasma as meat binders to give the desired shape in minced meat products. Collagen and its denatured form known as gelatin casing, is considered an important molecule. It offers excellent uniformity of appearance and strength and is usually used to fix the size and shape of processed meat products [84]. Gelatin solution is often injected into cured meats, which solidifies and increases water retention and resistance to cutting. Again, being a potential source of cheap and high-quality protein, chickpea flour added to comminuted meats increases cooking varieties and provides a softer texture that becomes commercially more valuable. Addition of organic compounds, such as melamine, milk, and urea, are also reported [85].

#### 4.2.2. Protein Substitution in Milk

Milk protein, which is considered a precursor of many bioactive peptides with antimicrobial activity, is not only a good source of calcium, zinc, copper, and phosphate ions but also helps in the absorption of many other nutrients [86,87,88]. Compounds with high nitrogen content can mimic a high protein concentration [88]. As the non-protein nitrogen cannot be distinguished by Kjeldhal and Dumas methods used for the determination of total protein content, the addition of various nitrogenous compounds in dairy products to increase the apparent protein content is quite frequent. In this case, melamine, urea, and whey are the most reported agents [88,89]. 

Addition of urea has also been reported several times. A low concentration of urea is naturally present in milk that generally comes from the grass or feed given to dairy cattle [45,90]. Due to being comparatively cheaper than other alternatives, it is extensively used to increase the apparent protein content [60,76]. If the amount of urea is found greater than the permissible limit of 10–16 mg/dL, it simply indicates an external addition [90]. 

Whey, being produced in large volume as a cheap byproduct of cheese and caseinates manufacturing, is added to liquid milk as whey protein to increase both the protein content and the total volume [60,76]. The fraudulent addition of rennet whey solids has been reported several times. As cheese whey costs four to five times less than milk and does not affect the sensory properties, it is a lucrative option for dishonest traders [91,92].

Vegetable proteins, such as low-grade soya powder, pea, and wheat also contribute to increasing the protein content of milk and are a convenient option due to being cheap and easily available. Rice and almond proteins are sold as milk supplements for consumers possessing lactose intolerance [15,93].

#### 4.2.3. Protein Substitution in Egg and Egg-Derived Foods

For eggs, melamine is a preferred compound to mimic proteins due to its nitrogen content of approximately 66.7%. Similarly, egg powder and some other egg-derived food products are also at risk of being contaminated with melamine to increase the apparent protein content for uplifting their market value [94].

#### 4.2.4. Protein Substitution in Staple Foods

Staple foods undergo various types of substitution involving quality discrimination. One of the most well-known substitution cases is the admixture of gluten containing cereals in gluten-free products. Gluten induces allergenic reactions to many individuals for whom gluten-free food has gained much popularity. However, many traders violate the regulation of compulsory declaration of the presence of elements prone to create allergenic reaction through the fraudulent admixing of gluten-containing cereals in gluten-free diets [95,96,97]. Peanuts are also another allergen mixed into bulk cereals, legumes, oilseeds, and bulk samples [98]. Lupin flour is also used as a soybean protein substitute [99]. In addition, wheat flour, gluten, and soybeans are frequently adulterated with melamine to increase their apparent protein content [96].

### 4.3. Substitution of Fat Content

#### 4.3.1. Fat Substitution in Milk

Being one of the major constituents of milk, the milk fat consisting of 97–98% triacylglycerols is present in all dairy products that contain milk and accounts for 3 to 5% in m/m of cow’s milk [60,100]. However, milk fat is costlier than most other edible fats; thus, production of milk derivatives is also an expensive process. Therefore, manufacturers often replace it with cheaper fats, such as vegetable fats or animal fats, not only for reducing manufacturing cost and to achieve an additional economic benefit but also to hide the effect of fraudulent dilution [15,76,101]. 

#### 4.3.2. Substitution of Oil and Fat Content in Oil and Fat

Due to higher market value and limited production, some oils are more at risk of being adulterated. Extra virgin olive oil is high quality olive oil [102,103]. It is an important source of fatty acids and natural antioxidants [104]. It has a high market value for its nutritional property, excellent taste, and aroma. However, the market price difference between extra virgin olive oil and other edible oils encourages the dishonest traders to adulterate it with other cheaper oils with similar fatty acid and sterol profiles. Hazelnut oil, seed oils, esterified oils, refined olive oils, and olive–pomace oils are examples of such oils [101,105,106]. Virgin coconut oil is another type of high-value oil, which is often adulterated with palm kernel oil, palm oil, and lard. Cod liver oil possesses some therapeutic effects on human health due to the presence of fatty acids and other nutritional values. It is usually adulterated with animal fats, especially lard, chicken fat, mutton fat, and beef fat [103]. Pure sesame oil, having a unique flavor and high nutrition, is adulterated with various cheaper oils, such as soybean oil, corn oil, rapeseed oil, and palm oil [106]. In this category, the other possible adulterations are adulteration of soybean oil to groundnut and sunflower seed oil, adulteration of sunflower seed oil to safflower oil, adulteration of borage oil to evening promise oil [107], adulteration of rice bran oil to mustard oil [107], and adulteration of argemone Mexican seed oil to other edible oils. Mineral oils, such as sesame oil, linseed oil, cottonseed oil, and castor oil are found in high-value oils. Minimally processed oils or cod-pressed oils are often replaced by refined oils. Adulteration of vegetable oil with lard and beef tallow to minimize production cost of margarines and shortenings is also reported [107].

### 4.4. Food Dilution

Food dilution is the addition of a cheaper ingredient to a high-value food stuff without declaring it [20]. Dilution and overdilution are widely practiced adulterations. Not only liquid but also solid food items are prone to dilution with water or other liquids. This phenomenon is conventionally confirmed by refractometric Brix determination or density measurements. It is the most common form of adulteration in liquid milk [45]. It affects the density, refractive index of lactoserum, and freezing point of adulterated milk items [76]. Honey and other sweeteners are also often diluted with water, which causes honey to deteriorate faster during storage. Therefore, honey overdiluted with water lacks proper consistency and nutritional value [37]. Meat products are also prone to such activity. The lean meat acquires a high water-binding capacity after being chopped and, thus, absorbs a large quantity of water, which has been claimed to give necessary consistency for stuffing into thin cases. It was detected frequently in frankfurters, bologna, and pork sausages [85]. Technology is used to fraudulently increase the weight of fish and seafood in order to make extra economic benefit. Over-treating, which means over-breeding or over-glazing, soaking fish in brine solution, injecting chemicals to increase muscle water-holding capacity, injection of fish byproducts back into the fillet, and water addition are such instances [14,64].

Fruit juices in high demand are reported to be substituted and adulterated using cheap fruit juice [49]. Orange juice is adulterated with other citrus juices when their prices fall [108]. Adulteration of pomegranate juice with natural grape pigments to represent the actual color is practiced. Adulteration with poor quality juice and peel extract is another well-known practice [109]. Commercialization of reconstituted juice made from concentrate as fresh squeezed juice is also practiced [49,110]. The admixing of vinegar derived from a C3 or C4 plant is widely accomplished due to economic advantage and availability [111]. Admixing various proportions of wine vinegars and alcohol vinegars to reduce production cost and selling as pure wine vinegar is another frequently reported malpractice [112].

Edible wheat consists of common wheat and durum wheat [113]. The coarse flour obtained from durum wheat is the primary ingredient of pasta. Pasta made from 100% durum wheat is considered of the highest quality since it imparts unique and firmer dough. Therefore, there is a price difference between these two types of wheats that exists in the market and tempts the manufacturers to admix common wheat with durum wheat [95,96,114]. Due to the growing interest in organic food, organic wheat flour has a huge market demand and is often adulterated with common wheat flour, cassava flour, and corn flour. Their visual detection is quite difficult, as organic wheat flour is nearly the same in color as cassava flour and corn flour [115].

Another widespread adulteration is the admixing of high-quality rice with low quality rice. Rice of different varieties are cultivated which can be admixed fraudulently during the cultivation, harvesting, transporting, and processing [116]. As most varieties of rice are almost similar to look at, their visual discrimination is nearly impossible [117]. Basmati rice, grown in India and Pakistan, holds a prime position among the more than 5000 rice varieties all over the world for its high quality and fragrance. It is sold at 2–3 times higher in price than other varieties in the market [118]. These phenomena provide an incentive for dishonest merchants to adulterate Basmati rice with non-Basmati rice, such as Jasmine rice, long grain rice, etc. [119].

Spices and spices powders are frequently subject to substitution with low quality substances [120]. Table 3 presents spices and their adulterants.

Dried bread, corn meal, potato starch, crackers, waste biscuit, and boiled rice are usually used as fillers in sausages [85]. Instead of the natural smoking of meat, smoke aromas are fraudulently used [71]. Other additives, such as natural bacon flavors, glycerin, and lecithin from animal fat, are also in use. Alcohol ingredients made from pork fat, such as lard, mono- and diglycerides, sodium stearoyl lactylate, and polysorbate 60 or 80, are also added. Grain or plant-based ingredients with pig-based carrier, such as Beta carotene, are also used [67]. Salt is used to increase the water holding capacity and weight of meat products artificially for economic gain, which causes changes in the secondary structure of the proteins [118].

The substitution adulteration with food items is summarized in Table 4.

## 5. Organic and Synthetic Adulterants

Illegal addition of organic acids, alcohols, and esters in certain food products are frequently reported. Synthetic pharmaceutical compounds and drugs are also added into food items to induce therapeutic effects. Dietary supplement is the most famous food category in this regard, making headlines in newspapers and food crime logs regularly.

### 5.1. Adulteration with Soymilk

Soymilk possesses similar properties to cow milk. Hence, soymilk is often fraudulently added to cow milk and buffalo milk in preparation of different dairy products [76,126]. The production cost of soymilk is 70% less than pure milk [15]. A combination of urea, vegetable oil, emulsifier, fat, and nitrogen content provide synthetic milk the same color, specific gravity, and consistency of pure buffalo milk and becomes undetectable. Its milky aroma turns it into a commercially valuable product; thus, 5–10% adulteration of dairy milk has been reported [76].

### 5.2. Adulteration with Organic Acids

Organic acids, such as malic acid, are added to apple juice concentrate to increase commercial value [45]. Unauthorized addition of organic acids, such as citric and tartaric acid, is beneficial because the sensory properties and commercial value of certain type of fruit juices rises with acidity level. Amino acids, such as glycine and glutamic acid or protein hydrolysates, are also added to food items to boost the total amino acid content. Addition of mixture of flavors, organic acids, and sugars is a commonly added chemical cocktail to fruit extract [11].

### 5.3. Adulteration with Synthetic Acetic Acids

The authenticity issues associated with vinegar is related to the raw material source and manufacturing processes [111]. Synthetic acetic acid is produced from non-biological origins obtained from either petroleum derivatives or by pyrolysis of wood. Synthetic acetic acid is reported to be sold as organic acetic acid or mixed with organic acetic acid to increase the volume [112,127,128].

### 5.4. Adulteration with Artificial Wine

Some adulterated wines are called artificial wine that consists of components organoleptically perceived as grape wine. Water, yeast, sugar, potassium tartarate, crystalline tartaric and citric acids, tannin, glycerol, ethanol, and ethyl esters of high fatty acids are the typical constituents of artificial wine [42]. Glycerol, diethylene glycol, citric acid, and semi-volatile additives, such as propylene glycol, sorbic acid, and benzoic acid, are also mixed for wine enhancement; other compounds, such as rectified alcohols, components of non-grape origin, and natural and synthetic flavor compounds, are also added [41,42,49].

### 5.5. Adulteration with Unrecorded and Surrogate Alcohol

Such alcohols represent alcoholic beverages that either do not possess an official registration in the jurisdiction, are manufactured illegally, or are consumed by cross-border trade. Alcohols that are not produced for human consumption, such as medicinal alcohol, disinfectant alcohol, denatured alcohol, synthetic alcohol, and other industrial alcohols, are often added to alcoholic beverages to increase their alcohol content [42].

### 5.6. Synthetic Pharmaceutical Adulterants

Addition of synthetic pharmaceutical ingredients in dietary supplements is a great concern of the present time. Dietary supplements marketed for various health benefits are fraudulently admixed with pharmaceutical compounds to boost the desired effect on the human body [129].

Various approved pharmaceutical drugs and their analogues, which are often very difficult to detect, have been found in food supplements advertised as a remedy for diseases [129]. The phosphodiesterase type 5 (PDE-5) inhibitors, such as sildenafil citrate, tadalafil hydrochloride, vardenafil hydrochloride, udenafil, mirodenafil hydrochloride, lodenafil carbonate, avanafil, and their unapproved designer analogues, are fraudulently added to herbal supplements. More exotic analogues synthesized by minor modifications to parent structures of approved PDE-5 inhibitors also have been added by traders to make their detection much more difficult [130,131]. In addition, adulteration with optical isomers of tadalafil has also been reported [131,132]. Up to the year of 2018, 80 synthetic PDE-5 inhibitors were found in herbal supplements among which, 62% of sildenafil, 26% of tadalafil, 9% of vardenafil, and 3% of others were reported [133]. Analgesics, such as paracetamol, antihistamines, theophylline, bromhexine, diazepam, chlordiazepoxide, glibenclamide, hydrochlorothiazide, aminopyrine, and phenytoin are frequently found in food supplements. Non-steroidal anti-inflammatory drugs, such as aspirin, mefenamic acid, and phenacetin, have also been reported in several dietary supplements [129,130,134,135,136,137].

To get rid of obesity and extra weight, people purchase several weight-reducing dietary supplements, which are often adulterated with synthetic drugs. Some are controlled by regulatory agencies and others are banned due to their adverse effect on human health [138,139]. Table 5 shows some examples of these adulterants.

Plant food supplements prepared for body-building and athletic performance enhancement are often found adulterated with anabolic steroids or prohormones. Performance-enhancing drugs listed by the World Anti-Doping Agency [140] are defined as pharmacological substances by World Anti-Doping Code and are not allowed for human therapeutic use. Some of those drugs are presented in Table 6.

Other related substances are growth hormone, erythropoietin, chorionic gonadotropin, β-2 agonists, hormones, and metabolic modulators, such as aromatase inhibitors and selective estrogen receptor modulators. Cannabinoids and glucocorticosteroids are also prohibited in athletic competition. New, modified, or “designer” steroids are of greatest concern because of the relatively little information available about their pharmacology and probable side effects. The presence of such adulterants poses a major risk for athletes since it results in a positive anti-doping control test, and the World Anti-Doping Agency does not justify whether it was deliberate or accidental doping [139,145].

## 6. Fraud and Mislabeling

Quality and origin of foodstuffs are of great concern to the consumers as food grown in certain regions and some special species have higher economic value due to their superior quality. Environmental pollution of the geographical origins is also another considerable factor in this regard [146]. Food from high quality batches, species, and cultures are not only often replaced with low quality products but also mislabeled deliberately by the traders for extra profit. Such malpractice violates customers’ rights, reduces the benefits of local cultivators, and creates unfair business competition [147]. Traceability is assured only by labelling and administrative documentation and, thus, is prone to frequent fraudulent practice. The evidence of such issues with fish, seafood, meat, processed meat, and staple foods is found mostly in scientific reports [63,71,117,148]. There are also reports on origin fraud and mislabeling of tea, fruit juice, vinegar, honey, and alcoholic beverages [41,45,47,149]. Expensive honey, such as pine, thyme, orange blossom, chestnut, heather, manuka, acacia, litchi, and linden, are frequently reported for mislabeling with respect to botanical origin [150,151]. The plant-based food “Fava Santorinis” having protected designation of origin (PDO) is often replaced with inferior yellow split peas [152]. ‘Eglouvis’ lentils cultivated in the Ionian Islands is often subject to origin fraud [153].

Misrepresentation of geographical and botanical origin is seen mostly in the case of olive oils and cocoa butter. The quality and constituents of olive oils vary with certain regional characteristics, surroundings, and manufacturing technique, holding different commercial values [104]. For example, some olive cultivars of Olea europaea L. are known to possess better quality because of breeding and selection strategies. Olive oils derived from those regions or cultivars are specified by labelling. Due to their high market value, they are often misrepresented [154,155,156]. Extra virgin olive oil has protected designation of origin (PDO) due to possessing a high content of monounsaturated fatty acids, vitamins, and antioxidants. Due to high commercial value, it is often mislabeled [157,158].

## 7. Health Impact

There is much historical evidence on the health hazard resulting from food adulteration. Some food adulterants do not participate in health degradation. Those only affect the nutritional parameter and reduce food quality. Species substitution of coffee products is such an example. Other adulterants are potential source of mild to severe illness. The financial loss associated with hospitalization and medication is also not negligible.

Consumption of fruits adulterated with ripening agents have been proved to be carcinogenic to the human body [25]. Other health issues, such as headache, dizziness, nausea, and kidney failure are also visible [32,34,159]. Similarly, artificial coloring agents have shown carcinogenic and genotoxic properties [9]. Artificial sweetening agents have been proved to be clastogenic and genotoxic [42]. Uterine cancer, exhaustion, and loss of energy have been found to be associated with the consumption of growth hormones [29]. Consumption of melamine with food products causes renal failure, kidney stones, and infection in urinary tract [15,88,160]. People consuming food preserved with formaldehyde have fallen victim to disturbances in the nervous system, kidney, liver, and lungs [54].

Substitution of food items can also be health hazardous. Fishery substitution may result into the consumption of illegal poisonous fish species that even cause death [14]. Substituted meat products may cause allergic reactions, diabetes, and cardiovascular diseases [72,161,162]. Evidence of massive death has been found from the consumption of Raki, a Turkish traditional aniseed-flavored distilled spirit [49]. Substitution of spices have resulted into intoxication, neurological, and gastrointestinal problems in children [163]; anaphylaxis, liver, and stomach problems have also been reported [9]. Species substitution in milk and dairy products may cause allergic reactions [164]. Oil substitution may result in gall bladder cancer, epidemic dropsy, glaucoma, loss of eyesight, paralysis, liver damage, and cardiac arrest [105,165].

Some notable food adulterants and their possible health effects are presented in Table 7.

## 8. Conclusions

Food adulteration has been a major global concern due to its impact on health and economy. For years, food items have been decorated with artificial colorants to attract consumers. Protection of external freshness through the addition of several preservatives have been frequently accomplished by food traders. Deliberate substitution of high-quality food with inferior food products has been a regularly practiced issue. Alteration of nutritional parameters through fraudulent substitution has also been prominent. Food is directly related to health, and any form of alteration in its natural composition should be prohibited. The opportunities existing in favor of food contamination should be analyzed. Lack of proper legislations and their strict application is one of the root causes of rapidly increasing adulteration of food.

Though the food officials from around the world are trying to combat adulteration of food, there still exists some limitations. Adulterated food is usually not identified until it shows a health hazard. In addition, many developing countries still lag in terms of food adulteration analysis techniques. Proper law enforcement and regular inspection of food quality can bring about drastic changes. In the modern time, the contribution of scientists and researchers on food adulteration detection and quantification technologies is highly appreciable. The documentation of ever-increasing fraudulent ideas and practices with food must be kept up to date to cope up with food crimes. Different forms of food adulteration and associated health impacts should be well-documented and analyzed. Considering the detrimental health impacts, food safety and quality assurance is an urgent necessity. Considering the prospects of a rapidly expanding global food market, the regulation of food quality should be of prime importance. In a nutshell, food adulteration is a broad concept and cannot be managed only by the policymakers and executors. The food manufacturers and sellers, along with the customers, should contribute to making their country a safe place to live.

## Figures and Tables

**Figure 1 foods-12-00199-f001:**
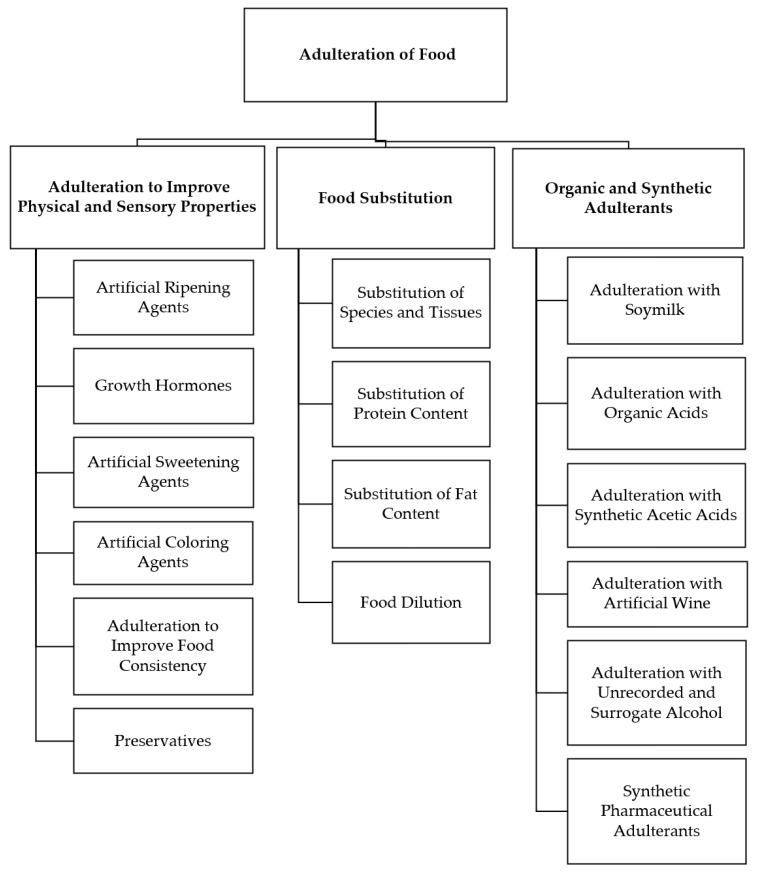
Major forms of food adulteration.

**Table 1 foods-12-00199-t001:** Ripening agents and their features.

Ripening Agent	Features
Ethylene	Treatment is accomplished either at the packing house or at the distribution place [32]. Climacteric fruits become ripened successfully with less than 1 ppm ethylene gas by volume with an exposure of 12 h or more [28,33]. The time as well as the required concentration for ripening may vary from 0.1 to 1 ppm depending on the commodity [34].
Ethanol [31]	Exogenously applied to inhibit ethylene biosynthesis for ripening purposes. A 35–50% ethanol treatment is effective against microbial and fungal attack, improves sensorial quality, and prevents table grape decay.
Ethylene Glycol [25]	Cheaper than ethylene gas. Usable by dilution with water. Addition of water may fasten the ripening process of various fruits in colder climactic conditions. Alkyl alcohol consisting of 6 and 14 carbon atoms can artificially ripe bananas. Lauryl alcohol can develop to a completely yellow color without affecting palatability within 48 h during treatment with 0.01% by weight of bananas.
Calcium Carbide [34,35]	Often used traditionally in granular or powder form. The high cost and scarcity of ethylene availability, faster ripening capability due to break down of glucose. and being comparatively cheaper than other chemicals are the root causes behind its vast usage. Sometimes, industrial grade calcium carbide shows impurities, such as Ca_3_P_2_, CaS, Ca_3_N_2_, SiC_2_, AsS_3_, and PH_3_. Acetylene is produced from the reaction of calcium carbide with moisture initiates ripening in mangoes at 1 mL/L and in bananas at 2.8 mL/L or 1 mL/L.
Ethephon [25]	Above pH 5, it decomposes into ethylene gas, bi-phosphate ion, and chloride ion. Provides better color profile than naturally ripened fruits and also improves the peel color of mangoes. Increases skin color, skin carotinoids, total suspended sloid, vitamin C, and fruit juice. Fruits treated with ethephon possess longer shelf life than fruits treated with calcium carbide, and it causes faster ripening. Some studies showed that pineapples, bananas, and tomatoes treated with 1000 ppm ethephon took less time for ripening than with other reagents.
Methyl Jasmonate [31]	Improves firmness, increases resistance to mechanical damage, and reduces microbial attack and post-harvest decay. Resists postharvest diseases, increases polyamine content, and elongates shelf life. Application of 0.1–10 mM jasmonate causes chlorophyll degradation, enhances carotene accumulation, and, thus, changes the color of apples and mangoes. A combined treatment of 0.1 mM methyl jasmonate and ethanol on strawberry shows higher antioxidant capacity, total phenolics, and anthocyanins than treatment with ethanol only.

**Table 2 foods-12-00199-t002:** Preservatives and corresponding food items [61,62].

Preservative	Food Item
Sodium benzoate (E211)	Carbonated drinks, pickles, sauces, and certain medicines
Sulfur dioxide (E220)	Carbonated drinks, dried fruit juices, cordials, and potato products
Sodium meta-bisulfite	Preservatives and antioxidants
Potassium nitrate (E249)	Cured meats and canned meat products
P-hydroxy benzoic acid esters (parabens)	Preserved foods and pharmaceuticals
Lactic acid bacteria	Fermented foods
Mono sodium glutamate (MSG)	All frozen foods, canned tuna, and vegetables
Sodium nitrite and sodium nitrate	Processed meats and fish to retain red color and avoid botulism
Trans fat	Deep-processed fast foods and certain processed foods
Sodium sulfite (E221)	Used in wine making and other processed foods
Potassium bromates	White flour, bread, and rolls
Propyl gallate and tertiary butyl hydroquinone	Processed foods, vegetable oils, and meat products

**Table 3 foods-12-00199-t003:** Spices and their adulterants [9,120,121,122].

Food Item	Adulterant
Chili	Oil, rice flour, bran, fruits, plant husks, rice powder, sawdust, talc powder, brick powder, and salt powder
Oregano	Sumac, olive leaves, myrtle leaves, *Satureja montana* L., *Origanum majorana* L., *Cistus incanus* L., *Rubus caesius* L., and *Rhus coriaria* L.
Cumin	Almond, peanut, treenuts, peach, cherry, fennel seeds, coriander seeds, caraway seeds, anise seeds, black cumin, and white cumin
Black Pepper	Chili, buckwheat or millet, powdered papaya seed, berries of wild pepper species, dried fruits of *Lantana camara*, *Embelia ribes*, seeds of *Mirabilis jalapa*, berries of *Schinus molle*, cheaper plant material of similar color, size, and shape, low-quality exhausted pepper, and stem and chalk of black pepper
Cinnamon and Nutmeg	Coffee husk
Chinese star anise	Japanese star anise
Paprika	Almond, white pepper, curcuma, barium sulphate, and brick powder
Saffron	Saffron of unknown origin, flower petals, and styles, old or deteriorated saffron materials, marigold leaves, arnica, beet, pomegranate fibers, dyed corn stigmas, cut or dyed *C. sativus* stamens, curcuma powder rhizomes, safflower, and calendula petals
Turmeric	Chalk powder, *Curcuma zedoaria,* and *Curcuma malabarica*
Garlic	Cornstarch
Sage	Ginger and onion powder and olive leaves
Black pepper berries	Mineral oil and dried papaya seed
Chili fruits	Mineral oil
Ginger	Lime, capsaicin, and exhausted ginger
Cardamom fruits	Small pebbles, orange seeds, and unroasted coffee seeds
Clove	Magnesium salt, sand, earth, and exhausted clove
Cinnamon powder	Eugenol, cylon oil, beechnut husk, hazel nut, and almond shell dust
Aniseed powder	Fennel

**Table 4 foods-12-00199-t004:** Various substitution adulteration with food items.

Substitution Issue	Food Category	Substituted Compound
Substitution of Species and Tissues	Fishery substitution	Crustaceans and high-quality shrimps, etc. [65]
	Substitution in meat products	Beef and horse meat, beef and mutton, poultry and pork, sausage, minced meat, etc. [67,68,85]
	Substitution in milk products	Cow milk, sheep milk, goat and buffalo milk, feta, manchego and pecorino cheese, etc. [77,123]
	Coffee substitution	Arabica and robusta coffee [79,80,81]
Substitution of Protein Content	Protein substitution in meat	Protein from animal or vegetable origin, enzymes, etc. [68,71,82]
	Protein substitution in milk	Melamine, urea, whey, vegetable proteins, etc. [45,88,91,93]
	Protein substitution in egg and egg-derived food	Melamine [94]
	Protein substitution in staple food	Gluten containing cereals in gluten-free products, peanut, lupin, etc. [95,98,99]
Substitution of Fat Content	Fat substitution in milk	Soybean oil, peanut oil, sunflower oil, coconut oil, beef tallow, pork lard, etc. [60,124,125]
	Substitution of oil and fat content in oil and fat	Extra virgin olive oil, virgin coconut oil, pure sesame oil, etc. [102,103,106]
	Fruit juice and vinegar	Cheap fruit juice or vegetable juice, reconstituted juice, vinegar derived from a C3 or C4 plant, wine vinegar, alcohol vinegar, etc. [49,111,112]
Substitution by Low Quality Food Products and Substances	Staple food	Common wheat and durum wheat, Basmati rice and non-Basmati rice [113,118]
	Spices	Oil, bran, fruits, plant husks, olive leaves, myrtle leaves, etc. [9,121]
	Substitution by other non-meat ingredients in meat products	Dried bread, corn meal, potato starch, smoke aroma, blood, natural bacon flavors, glycerin and lecithin from animal fat, lard, mono- and diglycerides, sodium stearoyl lactylate, etc. [67,71,85]

**Table 5 foods-12-00199-t005:** Adulterant category for psychiatric issues, obesity, and other health problems with examples [138,139].

Adulterant Category	Example
Anorexics	Amfepramone, rimonabant, fenproporex, phentermine, sibutramine, orlistat, mazindol, and fenfluramine
Anxiolytics or benzodiazepines	Diazepam, flurazepam, clonazepam, alprazolam, medazepam, midazolam, oxazepam, oxazolam, and chlordiazepoxide
Antidepressants	Fluoxetine, paroxetine, sertraline, bupropion, and diuretics, such as furosemide, spironolactone, and hydrochlorothiazide
Laxatives	Phenolphthalein and stimulants, such as ephedrine, norephedrine, and synephrine

**Table 6 foods-12-00199-t006:** Adulterant category to improve body-building and athletic performance with examples [139,140,141,142,143,144].

Adulterant Category	Example
Anabolic agents	AAS, clenbuterol, selective androgen receptor, modulators, tibolone, zeranol, zilpaterol, peptide hormones, and growth factors
Diuretics and certain masking agents	Acetazolamide, carmerone, indaparid, and plasma expanders
Stimulants	Amfepramone, meferox, and pseudoephedrine
Nacrotics	Buprenorphine, dextromoramide, methadone, morphine, and oxycodone
New, modified, or “designer” steroids	Prostanozol, methasterone, andostatrienedione, androstenedione, 5-androstern-3β-ol-17-one, methandienone, testosterone esters, androst-4-ene-3β-17β-diol, and boldenone

**Table 7 foods-12-00199-t007:** Food adulterants and their possible health effects.

Adulterant	Health Effect
Artificial ripening agents	Carcinogenic, headache, dizziness, mood disturbances, sleepiness, mental confusion, vomiting, diarrhea, permanent skin damage, kidney failure, etc. [25,32,34,159]
Artificial sweetening agents	Clastogenic, genotoxic, headache, dry mouth, nausea, vomiting, dizziness, DNA damage, and cancer [29,42,43]
Artificial coloring agents	Genotoxic, carcinogenic, multi-organ tissue injury, adverse effects on immune system and reproductive system, heart problems, mutagenicity, allergic reaction, hyperactivity, anemia, brain tumors, stomach diseases, brain injury, pigmented contact dermatitis, and stomach problems [9,29,166,167,168,169,170,171,172,173,174]
Preservatives	Disturbance in the nervous system, kidney, liver, and lungs [54]
Substituted meat products	Allergic reactions, diabetes, and cardiovascular diseases [72,161,162]
Substitution of spices	Intoxication, neurological, and gastrointestinal problems in children, anaphylaxis, and liver and stomach problems [9,163]
Oil substitution	Gall bladder cancer, epidemic dropsy, glaucoma, loss of eyesight, paralysis, liver damage, and cardiac arrest [105,165]

## Data Availability

Data is contained within the article.

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
