# Peer review of "Mechanisms and Health Aspects of Food Adulteration: A Comprehensive Review"

_foods, 2023, doi:10.3390/foods12010199_

Round 1

Reviewer 1 Report

In the present manuscript the authors propose a comprehensive review on intentional and unintentional food adulterations and the health impact linked to each adulteration/fraud. The paper is, overall, discretely written and structured. However, I do not consider the manuscript particularly innovative, there are many papers on this topic. A review on such a well-known topic must include recent references and up-to-date information. In this review many bibliographical references are outdated. This means that the literature survey carried out is limited and does not cover the last three years of scientific research on the topic (most recent references are dated 2019). The authors should reconsider their approach to research sources and try to produce an up-to-date paper with all the latest developments in this field. Furthermore, the criteria used to search for references should be clearly stated in the manuscript (dates, databases, keywords). Other specific suggestions are given below.

There is no temporal contextualisation of the various food frauds. What are the emerging adulterations in the last years? Which ones do we really need to worry about? Which frauds are most reported by the competent authorities? Use tools that can provide you data on the most commonly reported adulterations.

You provide a single paragraph on the public health impact of the various contaminants/adulterations. This paragraph provides only generic information without up-to-date information about the real health impact of each contaminant/adulteration. Updated references on actual and certified cases of human health impacts documented in recent years should be included in these sections. Also try to include the real impact that some contaminants might have in relation to new dietary trends, such as the consumption of seaweed, ethnic foods, unconventional foods, etc.

L33-34 It seems to me an imprecise definition. Adulteration of food is not only a question of change in nutritional properties, but there are other factors to consider as you have specified at several points in the paper.

L47-52 Include a reference to support the statement.

L83-86 Data from many years ago. Provide recent data on food-borne diseases.

L205-214 Colouring agents are only used in fruit, vegetables, egg, spices, and honey? Are there cases of fraud involving the addition of colouring agents in meat or fish?

L239-246 There are other substances besides formalin that are used fraudulently. In addition, please provide other details about formalin. Are there threshold values for formaldehyde? It is suggested to use recent references, as example:

·      Jinadasa, B. K. K. K., Elliott, C., & Jayasinghe, G. D. T. M. (2022). A review of the presence of formaldehyde in fish and seafood. Food Control, 108882.

L228-232 Include a reference to support the statement.

L287-343 Briefly describe the recent methods for detecting a substitution fraud for each food category.

Table 7. I think that should be appropriate to extend the analysis of pesticide to other countries besides Bangladesh.

L580-587 There is no mention of metal contamination in fish and other marine products (e.g. seaweed). I think it is very important to include data on this. There are many papers on metal contamination in seafood, as example:

·   Arulkumar, A., Nigariga, P., Paramasivam, S., & Rajaram, R. (2019). Metals accumulation in edible marine algae collected from Thondi coast of Palk Bay, Southeastern India. Chemosphere, 221, 856-862; Liu, Q., Liao, Y., & Shou, L. (2018).

·       Panebianco, F., Nava, V., Giarratana, F., Gervasi, T., & Cicero, N. (2021). Assessment of heavy-and semi-metals contamination in edible seaweed and dried fish sold in ethnic food stores on the Italian market. Journal of Food Composition and Analysis, 104, 104150;

·     Liu, Q., Liao, Y., & Shou, L. (2018). Concentration and potential health risk of heavy metals in seafoods collected from Sanmen Bay and its adjacent areas, China. Marine Pollution Bulletin, 131, 356-364.

L589-595 Too general. I do not know if it is appropriate to include a section on microbial contamination in a review on food adulteration. If you really want to provide this, more information should be given including all food categories and the main microbiological issues related to each food.

L617-630 Too generic. With regard to fish, for example, which families are histamine-risk?

In conclusion, in my opinion this manuscript needs a thorough revision before being reconsidered for publication. I hope I have provided you some input for improving the manuscript. I believe that a review on such an important topic should not be merely descriptive, but closely contextualised with the current historical period and updated with the latest news on the topic.

Author Response

Reviewer#1:

In the present manuscript the authors propose a comprehensive review on intentional and unintentional food adulterations and the health impact linked to each adulteration/fraud. The paper is, overall, discretely written and structured. However, I do not consider the manuscript particularly innovative, there are many papers on this topic. A review on such a well-known topic must include recent references and up-to-date information. In this review many bibliographical references are outdated. This means that the literature survey carried out is limited and does not cover the last three years of scientific research on the topic (most recent references are dated 2019). The authors should reconsider their approach to research sources and try to produce an up-to-date paper with all the latest developments in this field. Furthermore, the criteria used to search for references should be clearly stated in the manuscript (dates, databases, keywords). Other specific suggestions are given below.

Response to Reviewer: Noted. We have removed the outdated bibliographic references and added information from recent publications. The keywords used to search for references have been stated on page 3.

There is no temporal contextualisation of the various food frauds. What are the emerging adulterations in the last years? Which ones do we really need to worry about? Which frauds are most reported by the competent authorities? Use tools that can provide you data on the most commonly reported adulterations.

Response to Reviewer: Noted. We have added recent food adulteration incidents on page 2 and 3.

You provide a single paragraph on the public health impact of the various contaminants/adulterations. This paragraph provides only generic information without up-to-date information about the real health impact of each contaminant/adulteration. Updated references on actual and certified cases of human health impacts documented in recent years should be included in these sections. Also try to include the real impact that some contaminants might have in relation to new dietary trends, such as the consumption of seaweed, ethnic foods, unconventional foods, etc.

Response to Reviewer: Agreed and noted. We understand that health impact of food adulteration is a wide and important issue. To give this topic due priorities we are preparing a separate manuscript on health impact of food adulteration. Your suggestions would be highly useful for our upcoming manuscript. In this manuscript we provide a brief overview of health impacts to keep this manuscript concise and to avoid any repetition.

L33-34 It seems to me an imprecise definition. Adulteration of food is not only a question of change in nutritional properties, but there are other factors to consider as you have specified at several points in the paper.

Response to Reviewer: Agreed and Noted. We have revised and updated the definition of food adulteration on page 1.

L47-52 Include a reference to support the statement.

Response to Reviewer: Agreed and Noted. We have included reference no. 8 on page 2 to support the statement.

L83-86 Data from many years ago. Provide recent data on food-borne diseases.

Response to Reviewer: Agreed and Noted. We have included recent data on food-borne diseases on page 2 and 3.

L205-214 Colouring agents are only used in fruit, vegetables, egg, spices, and honey? Are there cases of fraud involving the addition of colouring agents in meat or fish?

Response to Reviewer: Noted. We have updated the information regarding colouring agents in food on page 10.

L239-246 There are other substances besides formalin that are used fraudulently. In addition, please provide other details about formalin. Are there threshold values for formaldehyde? It is suggested to use recent references, as example:

  • Jinadasa, B. K. K. K., Elliott, C., & Jayasinghe, G. D. T. M. (2022). A review of the presence of formaldehyde in fish and seafood. Food Control, 108882.

Response to Reviewer: Noted. We have added some details about formalin on page 12.

L228-232 Include a reference to support the statement.

Response to Reviewer: Noted. We have added reference 59 on page 11 to support the statement.

L287-343 Briefly describe the recent methods for detecting a substitution fraud for each food category.

Response to Reviewer: Noted. Detection of food adulteration and food substitution are not within the scope of the manuscript. To keep the manuscript brief and focused, we haven’t included detection methods and techniques in this manuscript.

Table 7. I think that should be appropriate to extend the analysis of pesticide to other countries besides Bangladesh.

Response to Reviewer: Agreed and noted. The authors have prior publications to reduce the above issues.

Ref 1: Khan and Rahman (2017) Pesticide Residue in Foods – Sources, Management, and Control https://doi.org/10.1007/978-3-319-52683-6

Ref 2: Debnath and Khan (2017). Health Concerns of Pesticides. In: Khan, M., Rahman, M. (eds) Pesticide Residue in Foods. Springer, Cham. https://doi.org/10.1007/978-3-319-52683-6_6

To keep this article concise and focused we have removed the table in the revised version.

L580-587 There is no mention of metal contamination in fish and other marine products (e.g. seaweed). I think it is very important to include data on this. There are many papers on metal contamination in seafood, as example:

Arulkumar, A., Nigariga, P., Paramasivam, S., & Rajaram, R. (2019). Metals accumulation in edible marine algae collected from Thondi coast of Palk Bay, Southeastern India. Chemosphere, 221, 856-862; Liu, Q., Liao, Y., & Shou, L. (2018).

Panebianco, F., Nava, V., Giarratana, F., Gervasi, T., & Cicero, N. (2021). Assessment of heavy-and semi-metals contamination in edible seaweed and dried fish sold in ethnic food stores on the Italian market. Journal of Food Composition and Analysis, 104, 104150;

Liu, Q., Liao, Y., & Shou, L. (2018). Concentration and potential health risk of heavy metals in seafoods collected from Sanmen Bay and its adjacent areas, China. Marine Pollution Bulletin, 131, 356-364.

Response to Reviewer: Agreed. We have mentioned the metal contamination in fish and other marine products on page 25 and 26.

L589-595 Too general. I do not know if it is appropriate to include a section on microbial contamination in a review on food adulteration. If you really want to provide this, more information should be given including all food categories and the main microbiological issues related to each food.

Response to Reviewer: Agreed and noted. Microbial contamination is a category of incidental food adulteration. We provided a brief overview of all the incidental food adulterants because all these types seem huge to discuss in detail.

L617-630 Too generic. With regard to fish, for example, which families are histamine-risk?

Response to Reviewer: Noted. We provided a brief overview of all the incidental food adulterants to keep the manuscript concise and focused.

In conclusion, in my opinion this manuscript needs a thorough revision before being reconsidered for publication. I hope I have provided you some input for improving the manuscript. I believe that a review on such an important topic should not be merely descriptive, but closely contextualised with the current historical period and updated with the latest news on the topic.

Response to Reviewer: Agreed and noted. This time, we tried to contextualize the manuscript with current historical period.

Reviewer 2 Report

This review is generally interesting for scientists working in the field (e.g. Food Scientists, Food Engineers, Analytical Chemists) and it can be published after some revision. In particular, a better grouping of the information can be provided.

In order to improve the quality of this review article, I have the following suggestions:

Some further kind of fraud should also be mentioned, such as:

§   Adulteration of fresh cow milk with reconstituted milk powder (e.g. in paragraph 4.1.3.) (https://doi.org/10.1016/j.talanta.2019.120223).

§   Mislabelling of honey in respect of its botanical origin (pine, thyme, orange blossom, chestnut, heather etc) and of course the most expensive honey, “Manuka” honey (https://doi.org/10.1016/j.vibspec.2022.103439, https://doi.org/10.1007/s13197-021-05181-7).

§   Mislabelling of various products in respect to their geographical origin. This case applies for famous product due to its geographical origin (e.g. foodstuffs with Protected Designation of Origin, such as “Fava Santorinis” (http://dx.doi.org/10.1016/j.foodchem.2016.06.088) and lentils “Eglouvis” (http://dx.doi.org/10.1017/S1479262118000096).

Olive oil is a product of high concern and I think some more information can be added (e.g. https://doi.org/10.1016/j.talanta.2020.120920, https://doi.org/10.1016/j.aca.2018.02.042).

Figure 1 and Page 5: “Dilution and overdilution” and “enhancement of food weight” have been written under the group of methods to improve physical and sensory properties. However, both dilution and increase of food weight are (similar) methods for adulteration and not to improve physical and sensory properties. They consist an individual category.

Figure 1, “Organic and Synthetic adulterants”: My recommendation is to provide general groups of adulterants (e.g. artificial sweetening agents, artificial coloring agents) without specifying the foodstuffs that are applied (e.g. milk, wine, etc).

I think a paragraph about the strategy of chemical analysis for the detection/ quantification of adulteration will be helpful for the reader. In brief, chemical analysis can be emphasized to specific analytes implying adulteration (e.g. melamine, antibiotics) or to provide a fingerprinting of the targeted original foodstuff (multivariate description of its chemical composition). In the latter case, we need to study a wide number of original (reference) samples and then to analyse the unknown samples. Comparison of the chemical description between the reference samples and unknown ones can be performed by means of chemometric techniques (e.g. class modelling, https://doi.org/10.1016/j.aca.2017.05.013). Some relevant reviews can be added, such as https://doi.org/10.1002/jsfa.8364

Conclusions: It should be extensively re-written. Do not duplicate text of the Abstract (e.g. “this article provides a picture of current food adulteration..”).  The section of conclusions is not like an abstract. Focus on conclusions, on the scope, on limitations and future perspectives.

Minor remarks:

Lines 79-83: I suggested authors use more recent information and update these data, as they reported in ref. 16 and 17, published in 2013 and 2014, respectively.

Lines 117-118: Some examples can be added.

Line 129: I suggest authors to remove the words “honey contamination from” because the use/ consumption of genetically modified plants is a more general problem.

Line 169: (i) Replace “coper” with “copper” and (ii) copper sulfate can not be considered as a growth hormone.

Lines 551-554: A better grouping of drugs is required. For example, aspirin, mefenamic acid and phenacetin are non-steroidal anti-inflammatory drugs.

Line 580 (and Table 1): Replace “heavy metal elements” with “heavy metals”.

Line 585: I have some reservations if Zn is a really toxic element.

Line 586: “Heavy metal and lead..”: Lead is also a heavy metal, isn’t it?

Lines 651-652: “..penicillin, …, β-lactams, …”: Penicillin is also a β-lactam antibiotic, why did authors distinguish penicillin from β-lactams?

“Trace and heavy elements” (e.g. Table 9): I suggest to replace it my “heavy metals and organometallic species”.

Author Response

This review is generally interesting for scientists working in the field (e.g. Food Scientists, Food Engineers, Analytical Chemists) and it can be published after some revision. In particular, a better grouping of the information can be provided.

In order to improve the quality of this review article, I have the following suggestions:

Some further kind of fraud should also be mentioned, such as:

Adulteration of fresh cow milk with reconstituted milk powder (e.g. in paragraph 4.1.3.) (https://doi.org/10.1016/j.talanta.2019.120223).

Response to Reviewer: Agreed and Noted. Supporting information has been added on page 14.

Mislabelling of honey in respect of its botanical origin (pine, thyme, orange blossom, chestnut, heather etc) and of course the most expensive honey, “Manuka” honey (https://doi.org/10.1016/j.vibspec.2022.103439, https://doi.org/10.1007/s13197-021-05181-7).

Response to Reviewer: Agreed and Noted. We have added information on page 25.

Mislabelling of various products in respect to their geographical origin. This case applies for famous product due to its geographical origin (e.g. foodstuffs with Protected Designation of Origin, such as “Fava Santorinis” (http://dx.doi.org/10.1016/j.foodchem.2016.06.088) and lentils “Eglouvis” (http://dx.doi.org/10.1017/S1479262118000096).

Response to Reviewer: Agreed and Noted. We have added information on page 25.

Olive oil is a product of high concern and I think some more information can be added (e.g. https://doi.org/10.1016/j.talanta.2020.120920, https://doi.org/10.1016/j.aca.2018.02.042).

Response to Reviewer: Agreed and Noted. We have added information on page 25.

Figure 1 and Page 5: “Dilution and overdilution” and “enhancement of food weight” have been written under the group of methods to improve physical and sensory properties. However, both dilution and increase of food weight are (similar) methods for adulteration and not to improve physical and sensory properties. They consist an individual category.

Response to Reviewer: Agreed and noted. We have included a separate section “Food dilution” and addressed both dilution and enhancement of food weight in that section on page 17.

Figure 1, “Organic and Synthetic adulterants”: My recommendation is to provide general groups of adulterants (e.g. artificial sweetening agents, artificial coloring agents) without specifying the foodstuffs that are applied (e.g. milk, wine, etc).

Response to Reviewer: This comment is not clear to the authors. The comment may not be relevant for this section of the manuscript. Further clarification of the comment would be highly helpful.

I think a paragraph about the strategy of chemical analysis for the detection/ quantification of adulteration will be helpful for the reader. In brief, chemical analysis can be emphasized to specific analytes implying adulteration (e.g. melamine, antibiotics) or to provide a fingerprinting of the targeted original foodstuff (multivariate description of its chemical composition). In the latter case, we need to study a wide number of original (reference) samples and then to analyse the unknown samples. Comparison of the chemical description between the reference samples and unknown ones can be performed by means of chemometric techniques (e.g. class modelling, https://doi.org/10.1016/j.aca.2017.05.013). Some relevant reviews can be added, such as https://doi.org/10.1002/jsfa.8364

Response to Reviewer: Noted. Detection and quantification of food adulteration are not within the scope of the manuscript. To keep the manuscript brief and focused, we haven’t included detection chemical analysis for the detection/quantificaito techniques in this manuscript.

Conclusions: It should be extensively re-written. Do not duplicate text of the Abstract (e.g. “this article provides a picture of current food adulteration..”).  The section of conclusions is not like an abstract. Focus on conclusions, on the scope, on limitations and future perspectives.

Response to Reviewer: Agreed and noted. We have rewritten the conclusion focusing on the scope, limitations and future prospects.

Minor remarks:

Lines 79-83: I suggested authors use more recent information and update these data, as they reported in ref. 16 and 17, published in 2013 and 2014, respectively.

Response to Reviewer: Agreed and noted. We have added recent information and data on page 2 and 3.

Line 129: I suggest authors to remove the words “honey contamination from” because the use/ consumption of genetically modified plants is a more general problem.

Response to Reviewer: Agreed and noted. We removed “honey contamination from genetically modified plants” from the manuscript.

Line 169: (i) Replace “coper” with “copper” and (ii) copper sulfate can not be considered as a growth hormone.

Response to Reviewer: Agreed and noted. Replaced “coper” with “copper”. Removed copper sulfate on page 9.

Lines 551-554: A better grouping of drugs is required. For example, aspirin, mefenamic acid and phenacetin are non-steroidal anti-inflammatory drugs.

Response to Reviewer: Agreed and noted. We have made a better grouping of drugs on page 22 and 23.

Line 580 (and Table 1): Replace “heavy metal elements” with “heavy metals”.

Response to Reviewer: Agreed and noted. We have replaced “heavy metals elements” with “heavy metals” on page 25 and 26.

Line 585: I have some reservations if Zn is a really toxic element.

Response to Reviewer: Agreed. Zn has been removed on page 25 and 26.

Line 586: “Heavy metal and lead..”: Lead is also a heavy metal, isn’t it?

Response to Reviewer: Agreed. Lead has been removed on page 25 and 26.

Lines 651-652: “..penicillin, …, β-lactams, …”: Penicillin is also a β-lactam antibiotic, why did authors distinguish penicillin from β-lactams?

Response to Reviewer: Agreed. We have removed β-lactam on page 28.

“Trace and heavy elements” (e.g. Table 9): I suggest to replace it my “heavy metals and organometallic species”.

Response to Reviewer: Agreed. We have made correction on page 25 and 26.

Reviewer 3 Report

This review paper is a mixture of various issues and topics – food fraud, food defense, food security, food safety and food quality. In parallel, references for defining some of these issues are from authors and journals not recognized as leading ones in these field (and wrong). Finally, incidental food adulteration associated with contaminants that cause food safety are not considered as food fraud, but as food safety outbreaks.

Authors need to know that there are scientific and international recognized definitions of food fraud (types, impacts etc.) which have not been analyzed by the authors. Authors need to revise the paper, clearly define food adulteration (in line with latest papers and recommendation from World Health Organization, US FDA, European Food Safety Authority, etc) and rewrite the entire review (omitting some parts that are totally irrelevant).

Author Response

Reviewer#3:

This review paper is a mixture of various issues and topics – food fraud, food defense, food security, food safety and food quality. In parallel, references for defining some of these issues are from authors and journals not recognized as leading ones in these field (and wrong). Finally, incidental food adulteration associated with contaminants that cause food safety are not considered as food fraud, but as food safety outbreaks.

Authors need to know that there are scientific and international recognized definitions of food fraud (types, impacts etc.) which have not been analyzed by the authors. Authors need to revise the paper, clearly define food adulteration (in line with latest papers and recommendation from World Health Organization, US FDA, European Food Safety Authority, etc) and rewrite the entire review (omitting some parts that are totally irrelevant).

Response to Reviewer: Noted and Thanks. We have revised the article, re-written different sections, removed old and insignificant references, added recent and more relevant references, and changed the title of the article.

To define food adulteration, the World Health Organization (WHO) mainly focuses on deliberate adulteration. WHO refers unintentional adulteration of food as food safety incidents. [Ref: https://www.who.int/news-room/fact-sheets/detail/food-safety]

In broad sense, food adulteration is referred to a process in which the quality of food is lowered. According to the US Food and Drug Administration (FDA), “Food can be declared adulterated if: i) a substance is added which is injurious to health, ii) cheaper or inferior quality item added to food, iii) any valuable constituent is extracted from main food article, iv) quality of food is below the standards, v) any substance has been added to increase bulk or weight

  1. vi) to make it appear more valuable”.

Many authors mention three types of food adulteration: I.              Intentional adulteration, II. Metallic contamination, III.              Incidental adulteration. In this manuscript, both Metallic contamination (II) and Incidental adulteration (III) are considered under the same category (Unintentional adulteration), since each adulteration involves food quality degradation by unintentional or accidental means.

Ref. 1. Food adulteration: sources, health risks and detection methods, by Sangita Bansal, Apoorva Singh, Manisha Mangal, Anupam K Mangal and Sanjiv Kumar,

DOI: 10.1080/10408398.2014.967834

Ref. 2. An overview of food adulteration: Concept, sources, impact, challenges and detection, by Ankita Choudhary, Neeraj Gupta, Fozia Hameed and Skarma Choton, DOI: 10.22271/chemi.2020.v8.i1am.8655

Ref. 3. Chapter 1 - Food Safety and Quality, by Shyam Narayan Jha, DOI: 10.1016/B978-0-12-420084 5.00001-9

The goal of our study is to aggregate all types of food adulteration in a single frame. Addressing the argument between food adulteration and food safety, we modified the title of our manuscript as, “Mechanisms and Health Aspects of Food Adulteration and Food Safety: A Comprehensive Review”.

“Origin fraud and mislabelling” is a form of food fraud, hence, we addressed it as food fraud.

Round 2

Reviewer 1 Report

The authors addressed all the observations improving considerably the manuscript. I believe the work is ready for publication. As a final remark, I only suggest to include in the introduction the databases where you searched for the mentioned keywords.

Author Response

Reviewer 1:

-The authors addressed all the observations improving considerably the manuscript. I believe the work is ready for publication. As a final remark, I only suggest to include in the introduction the databases where you searched for the mentioned keywords.

Response to Reviewer: Agreed. We have addressed the database in pages 2 and 3.